# Rethinking the High-throughput LLM Inference: An Opportunity for Speculative Decoding

## Abstract

Speculative decoding is a widely adopted method for accelerating autoregressive generation by drafting multiple candidate tokens and verifying them jointly with the target model. While effective in small-batch settings, it has been considered impractical under large-batch inference due to the belief that such regimes are compute-bound. Motivated by recent system-level findings that memory bandwidth, not compute, remains the dominant bottleneck in large-batch inference, we revisit the feasibility of speculative decoding under high-throughput conditions. We introduce $\gamma$-*tolerance*, a latency-based criterion that characterizes when speculative decoding provides tangible speedup, and empirically validate that acceleration remains attainable across practical batch sizes and system configurations. Building on this insight, we derive a revised success condition for speculative decoding and demonstrate that most existing drafter architectures violate it due to poor trade-offs between accuracy and efficiency. To address this, we identify Multi-Token Prediction with Gated LoRA as a promising approach and develop a high-performance implementation. Our system achieves up to $2.37\times$ speedup at batch size 256 without requiring long-context prompts or architectural changes to the target model, demonstrating that speculative decoding can be both feasible and effective in large-batch inference.

## 1 Introduction

The recent proliferation of powerful reasoning models, from proprietary systems like GPT-5 (OpenAI, 2025) and Gemini-2.5 (Comanici et al., 2025) to open-source alternatives such as Qwen3 (Yang et al., 2025) and DeepSeek-R1 family (Guo et al., 2025), is significantly expanding the applicability of large language models (LLMs). As these models enable a new class of complex applications, such as autonomous agents and sophisticated code generation (Ferrag et al., 2025; Prabhakar et al., 2025; Wei et al., 2025), the demand for serving infrastructure has grown significantly. Consequently, accelerating LLM inference has become more critical and urgent than ever before, as it is essential for both enhancing user experience and ensuring the economic viability of service providers.

Among various acceleration strategies, speculative decoding (Leviathan et al., 2023; Chen et al., 2023; Li et al., 2024a) stands out as a particularly promising direction. It leverages a lightweight *drafter* model to propose multiple candidate tokens, which are then simultaneously verified by the larger *target* model in a single forward pass. In contrast to compression-based techniques such as quantization or pruning (Frantar et al., 2023; Sun et al., 2024), this method leaves the architecture and parameters of the target model entirely unchanged. As a result, it enables a form of *lossless acceleration* that preserves the model's full capabilities, which is a critical requirement for high-fidelity applications involving complex inference or problem solving. However, despite this appealing property, speculative decoding has remained limited to small-batch inference (Cai et al., 2024; Li et al., 2024a; Xia et al., 2025), largely due to the prevailing assumption that large-batch settings are compute-bound, where the benefits of parallel verification are presumed negligible.

However, recent system-level analysis (Recasens et al., 2025) challenges the conventional belief that large-batch LLM inference is compute-bound, revealing that memory bandwidth remains the primary bottleneck even under well-optimized inference stacks. This finding undermines the core premise

that has long rendered speculative decoding ineffective in high-throughput scenarios, and reopens a fundamental question:

*Can speculative decoding remain effective under high-throughput, large-batch inference?*

To address this, we introduce a latency-based metric, $\gamma$-*tolerance*, which quantifies how the target model's forward latency scales with the number of tokens verified in a single forward pass. This formalism provides a precise and actionable condition for determining when speculative decoding is feasible. Our empirical analysis shows that although speculative decoding remains theoretically feasible in large-batch inference, the available headroom for acceleration is significantly narrower than in small-batch settings.

Recognizing the limited margin for acceleration necessitates re-evaluating the design principles that have guided speculative decoding in small-batch settings. While previous methods relied on generating many candidate tokens at low cost and deferring accuracy to the target model, we demonstrate that such *breadth-based* strategies collapse under large-batch conditions, where the verification cost scales with input length and leads to disproportionately higher latency. Instead, acceleration hinges on *accuracy-driven* drafting, where fewer but more reliable candidates are proposed and efficiently verified.

Guided by this insight, we revisit widely adopted drafter architectures and evaluate their suitability under large-batch constraints. We find that most existing approaches fall short of the accuracy–efficiency trade-off required for effective acceleration in large-batch inference. In response, we identify an alternative strategy based on *Multi-Token Prediction with Gated LoRA* (Samragh et al., 2025). Although their primary objective was not speculative decoding, this method aligns well with the requirements of large-batch speculative decoding due to its high draft accuracy and integrated design. By incorporating it into a high-throughput inference engine, we demonstrate that speculative decoding can achieve substantial acceleration even at large batch sizes, without relying on favorable conditions such as long-context inputs. Our contributions are summarized as follows:

- We introduce $\gamma$-*tolerance*, a latency-based criterion that formalizes the feasibility of speculative decoding in large-batch regimes. Through an empirical study, we validate that non-trivial speedups remain attainable under realistic conditions.

- We derive a revised success formula for achieving acceleration in speculative decoding and demonstrate that most existing drafter architectures violate this condition, highlighting the limitations of breadth-oriented strategies in large-batch settings.

- We identify a promising approach based on Multi-Token Prediction with Gated LoRA, and develop a high-performance implementation that attains up to $2.37\times$ speedup at batch size 256, without requiring favorable conditions such as long-context prompts.

## 2 RELATED WORK

**Speculative decoding.** Speculative decoding accelerates autoregressive generation by having a lightweight drafter propose multiple candidate tokens that a target model verifies in a single forward pass. A large body of work has extensively explored a variety of drafter instantiations, including standalone draft models (Leviathan et al., 2023; Chen et al., 2023), predictive-head or multi-branch heads that attach lightweight predictors to intermediate layers (Cai et al., 2024; Ankner et al., 2024; Li et al., 2024a; Zhang et al., 2025; Li et al., 2025a), and self-speculative approaches that reuse pruned, quantized, or partially executed variants of the target (Elhoushi et al., 2024; Zhang et al., 2024; Sadhukhan et al., 2025; Xia et al., 2025; Tiwari et al., 2025). Predictive-head methods are often paired with tree drafting (Miao et al., 2024) to compensate for lower draft accuracy (induced by ultra-lightweighting) by proposing many candidates.

More recently, large-batch regimes have been studied. TurboSpec (Liu et al., 2024) introduces a runtime controller for speculative decoding that adapts the number of proposed and verified tokens based on an online goodput objective, and evaluates the approach across various batch sizes and workloads. As such, it is best viewed as a system-level serving optimization rather than a redesign of the algorithm. MagicDec (Sadhukhan et al., 2025) analyzes that speculative decoding can provide speedup at high throughput for long-context inference and proposes a drafting strategy that leverages

compressed KV cache. Because its mechanism is built on KV cache compression, the method is primarily effective in long-context regimes, and its gains do not directly generalize to short- or medium-context workloads. Accordingly, its scope is complementary to ours, which targets high-throughput large-batch inference without assuming long contexts.

**LLM serving systems and kernels.** High-throughput serving has advanced via continuous batching and memory-aware schedulers (e.g., vLLM, SGLang) (Kwon et al., 2023; Zheng et al., 2024), together with IO-aware attention kernels. FlashAttention and FlashAttention-2 reduce HBM traffic through tiling and fusion and improve throughput via better parallelism and work partitioning (Dao et al., 2022; Dao, 2024). FlashInfer (Ye et al., 2025) targets workloads with a customizable attention engine that supports composable or block-sparse KV formats, JIT-fused kernels, persistent kernels compatible with CUDA Graphs, and load-balanced scheduling; empirically, it reports lower inter-token latency and higher bandwidth utilization than FlashAttention baselines in representative decode workloads.

Recent system profiling revisits the view that large-batch decoding becomes compute-bound. Recasens et al. (2025) use Nsight-based roofline and stall analysis to demonstrate that attention remains memory-bound at large batch sizes: DRAM bandwidth saturates, and stalls are dominated by memory; furthermore, higher matrix multiplication (MatMul) intensity does not shift the bottleneck. Comparing kernels (e.g., xFormers (Lefaudeux et al., 2022) versus FlashAttention) highlights kernel-dependent behavior: stall mix, cache use, and per-token latency scaling differ across kernels, indicating that kernel design and memory-access patterns materially affect effective bandwidth, even though decoding remains memory-bound at scale.

## 3 REVISITING THE FEASIBILITY OF SPECULATIVE DECODING FOR LARGE BATCH LLM INFERENCE

This section establishes the analytical and empirical basis for large batch speculative decoding. We first derive the necessary condition for speedup in terms of $\gamma$-*tolerance*, a measure of how target-model latency responds to longer verification inputs. We then present empirical measurements across batch sizes, sequence lengths, and attention kernels to evaluate whether this condition holds in practice.

### 3.1 THE NECESSARY CONDITION FOR SPECULATIVE DECODING

Speculative decoding accelerates autoregressive generation by employing a lightweight drafter to propose multiple future tokens that the target model verifies in a single pass, a process referred to as *parallel verification*. The efficiency of this mechanism is determined by how latency scales when the target model processes multiple tokens within a single forward pass. If latency remains nearly constant, several draft tokens can be verified with negligible additional cost, thereby reducing the number of sequential target invocations, which yields substantial acceleration. In contrast, if latency increases approximately in proportion to the number of tokens forwarded simultaneously, then verifying $\gamma$ tokens in one pass incurs nearly the same cost as generating them individually, and the potential gain largely vanishes.

This contrast elucidates why speculative decoding has conventionally been regarded as effective in memory-bound regimes but ineffective in compute-bound regimes. In the former case, latency is dominated by memory transfer (e.g., parameter loading), so extending the verification length introduces minimal marginal cost. In the latter, arithmetic operations dominate, and the computational cost scales almost linearly with the number of tokens, eliminating the benefit of parallel verification. In essence, the decisive factor is not merely whether the system is broadly classified as memory- or compute-bound, but rather the extent to which latency amortization occurs as the verification length increases. Capturing this scaling behavior is therefore essential for determining when speculative decoding can provide practical acceleration.

We now formalize this decisive factor by introducing a latency-based metric, $\gamma$-**tolerance**, which quantifies how latency scales with verification length:

$$\tau(\gamma) := \frac{T_{\mathcal{T}}(1)}{T_{\mathcal{T}}(\gamma)},$$

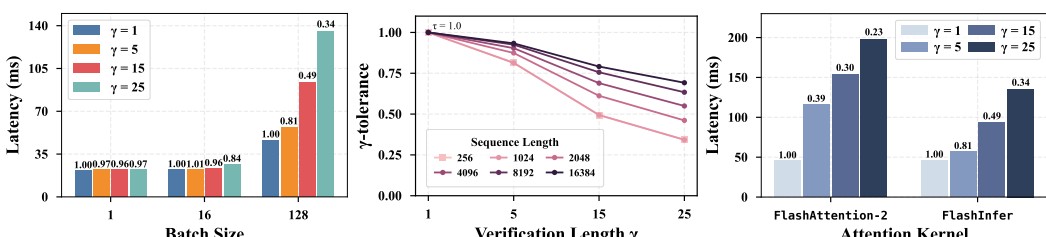

Figure 1: Empirical $\gamma$-tolerance across batch sizes, sequence lengths, and kernels. *(Left)*: $\tau(\gamma)$ and forward latency for $\gamma \in \{1, 5, 15, 25\}$ at batch sizes 1, 16, and 128 (sequence length fixed to 1024). *(Middle)*: $\tau(\gamma)$ as a function of past sequence length (KV cache size) at batch size 128. *(Right)*: Comparison between FlashAttention-2 and FlashInfer in terms of latency scaling and $\tau(\gamma)$ at batch size 128 and sequence length 1024. Numbers above bars indicate $\tau(\gamma)$.

where $T_{\mathcal{T}}(\gamma)$ denotes the latency of the target forward pass with verification length $\gamma$. Intuitively, $\tau(\gamma)$ measures how slowly latency increases as the verification length grows. A value close to 1 indicates that latency is nearly invariant to input length (highly memory-bound), while a value near $1/\gamma$ corresponds to almost linear scaling (compute-bound). In this sense, $\tau(\gamma)$ provides a concrete, measurable proxy for the classical memory/compute distinction and serves as a direct indicator of speculative decoding's potential.

With this notion, the condition for speedup can be precisely characterized. Let $\lambda$ denote the expected number of tokens generated per speculative step (the number of accepted tokens plus the bonus token), and let $T_{\mathcal{D}}$ be the drafter's per-step latency. Then, one speculative step costs $T_{\mathcal{T}}(\gamma + 1) + \gamma T_{\mathcal{D}}$, whereas sequential generation of $\lambda$ tokens costs $\lambda T_{\mathcal{T}}(1)$. The resulting speedup is

$$S = \frac{\lambda T_{\mathcal{T}}(1)}{T_{\mathcal{T}}(\gamma + 1) + \gamma T_{\mathcal{D}}} = \frac{\lambda \tau(\gamma + 1)}{1 + \gamma \delta \tau(\gamma + 1)}, \tag{1}$$

where $\delta = T_{\mathcal{D}}/T_{\mathcal{T}}(1)$. In the idealized case $T_{\mathcal{D}} \to 0$, this simplifies to $S \approx \lambda \tau(\gamma + 1)$, implying that a necessary condition for acceleration is $\tau(\gamma + 1) > 1/\lambda$. As $\delta$ grows, the requirement becomes stricter. Thus, $\tau(\gamma)$ bridges system-level intuitions with a formal performance metric: it not only reflects whether a model is operating closer to the memory- or compute-bound regime, but also provides a direct and testable criterion for when speculative decoding can yield tangible acceleration.

### 3.2 EMPIRICAL VALIDATION OF THE NECESSARY CONDITION

We next evaluate whether the feasibility condition is realized in practice under large-batch inference. Measurements are conducted on Qwen3-8B (Yang et al., 2025) with a single NVIDIA H100 GPU, using 1000 forward passes and 100 warmup runs to remove compilation and CUDA graph effects. Unless otherwise specified, the FlashInfer (Ye et al., 2025) kernel is employed, as it is widely adopted by modern inference engines, including vLLM (Kwon et al., 2023), SGLang (Zheng et al., 2024), and TensorRT-LLM (NVIDIA, 2025).

**Scaling with batch size.** Fig. 1 (left) shows that speculative decoding remains feasible even under large-batch inference, although the margin for acceleration narrows. For a chain draft of length 4 ($\gamma = 5$), we observe $\tau(5) = 0.81$ at batch size 128, corresponding to an upper bound of $S \approx 0.81\lambda$; with $\lambda \approx 3$ (approximately two accepted tokens plus the bonus token), this yields a $2.43\times$ theoretical upper bound of speedup. By contrast, deeper verification quickly undermines feasibility: at $\gamma = 25$, $\tau(25) = 0.34$, which caps the speedup near $1\times$ for $\lambda = 3$. More generally, as the batch size increases, the marginal latency per additional verified token rises, and the latency-$\gamma$ curve steepens; consequently, $\tau(\gamma)$ declines with the batch size, shrinking the feasible region. For example, at $\gamma = 5$, $\tau$ moves from $0.97$ (batch 1) to $0.81$ and at $\gamma = 25$, it drops from $0.97$ (batch 1) to $0.34$ (batch 128). Hence, under large batches, shallow drafts can still be effective, but the necessary condition $\tau > 1/\lambda$ is satisfied only for relatively small $\gamma$, and the margin for attainable acceleration contracts as batch size grows.

**Scaling with context length.** Fig. 1 (middle) quantifies how past context modulates $\gamma$-tolerance at batch size 128. In our measurements, $\tau(\gamma)$ generally increases with sequence length and exhibits

sublinear growth that tends to saturate beyond roughly 1k to 4k tokens, consistent with a shift toward memory-dominated execution. This shift materially relaxes the feasibility condition: for shallow verification ($\gamma = 5$), $\tau(5)$ exceeds the $1/\lambda$ threshold for typical acceptance levels (e.g., $\lambda \in [2, 3]$) across much of the measured range, whereas for deep verification ($\gamma = 25$) the threshold is only approached and, under our setup, not surpassed. The ordering $\tau(1) > \tau(5) > \tau(15) > \tau(25)$ is preserved across lengths, but the gaps shrink as context grows, indicating that longer prefixes preferentially enhance larger-$\gamma$ settings without fully closing the feasibility gap. Practically, workloads with sustained context (e.g., long-context) already operate in the regime where shallow drafts are justified under large-batch inference (Sadhukhan et al., 2025), while very deep drafts remain impractical even at the longest prefixes evaluated.

**Kernel dependence.** Fig. 1 (right) contrasts latency-$\gamma$ scaling across kernels at batch size 128 and sequence length 1024. With FlashAttention-2 (Dao, 2024), forward latency rises sharply already between $\gamma = 1$ and 5, depressing $\tau(\gamma)$ at small $\gamma$ and making the condition $\tau > 1/\lambda$ difficult to satisfy. By contrast, FlashInfer exhibits milder growth over the same range, yielding consistently higher $\tau(\gamma)$ and preserving feasibility for shallow verification (e.g., $\gamma = 5$). These differences are consistent with implementation-level effects—reduced launch overheads, more aggressive operator fusion, persistent execution, and cache-friendlier memory access patterns—leading to gentler latency-$\gamma$ scaling under FlashInfer. Consequently, feasibility under large-batch inference is as much a property of the serving stack as of the model.

Importantly, this kernel sensitivity plausibly accounts for prior empirical reports that speculative decoding was ineffective at large batch sizes: when evaluated atop kernels with steeper latency–$\gamma$ scaling (e.g., FlashAttention-2 in our measurements), $\tau(\gamma)$ at small $\gamma$ often falls below the necessary threshold, yielding $S \leq 1$ even for shallow drafts; with kernels exhibiting gentler scaling (e.g., FlashInfer), the same configuration can satisfy the feasibility condition and produce speedup. This reconciles seemingly conflicting observations in the literature.

## 4 FROM POTENTIAL TO PRACTICAL ACCELERATION

Section 3 established that speculative decoding can remain feasible in large-batch LLM inference, but only under much stricter conditions than in small-batch settings. We now turn from feasibility to practicality: *under what circumstances can speculative decoding deliver tangible speedup, and how do existing drafter architectures align with these requirements?* This section refines the success formula for speculative decoding in the large-batch regime, evaluates representative drafter families against the updated criteria, and finally identifies a promising work that reconciles accuracy and efficiency.

### 4.1 REVISING THE SUCCESS FORMULA FOR LARGE-BATCH INFERENCE

In small-batch inference, the dominant recipe for achieving strong speedups has been to employ an extremely lightweight drafter, generate many candidates per step through tree drafting, and rely on the target model to filter them in a single verification pass. This strategy proved effective because verification cost was nearly invariant with respect to input length; in practice, $\tau(\gamma) \approx 1$ even for large $\gamma$, so the target model could verify dozens of draft tokens at almost the same latency as one. From the perspective of $\gamma$-tolerance, this explains why even low-accuracy drafters could deliver strong speedups: breadth was inexpensive, and speculative decoding could increase $\lambda$ without incurring meaningful overhead. Formally, when $\tau(\gamma) \approx 1$, the speedup expression in equation 3.1 simplifies to $S \approx \frac{\lambda}{1+\gamma\delta}$. In the ideal case of $\delta \to 0$, this reduces to $S \approx \lambda$, meaning that larger trees (higher $\gamma$) directly increased speedup by raising $\lambda$. This is the basis for the breadth-oriented strategies that have dominated speculative decoding research in small-batch regimes.

However, this success formula does not carry over to large-batch inference. As shown in Section 3, $\gamma$-tolerance deteriorates rapidly with verification length: for example, $\tau(5) = 0.81$ and $\tau(25) = 0.34$ under realistic batch and sequence length configurations. This implies that the latency of the target model grows significantly with verification length, undermining the assumption that verification is nearly cost-free. Substituting such values into equation 3.1 illustrates the collapse: even with $\delta \to 0$, $\lambda = 3$ and $\tau(25) = 0.34$ yields $S \approx 1.0$, i.e., at best break-even rather than a gain. In practice, with

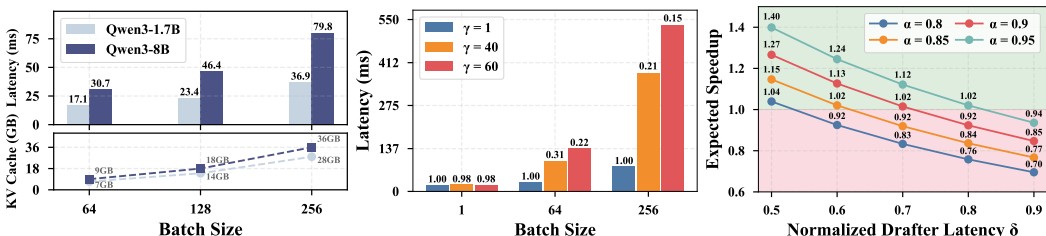

Figure 2: Empirical factors governing drafter viability under large-batch inference. *(Left)*: Forward latency and KV cache footprint of a standalone drafter (Qwen3-1.7B) vs. target (Qwen3-8B) across batch sizes. *(Middle)*: Effect of verification length and batch size on latency. *(Right)*: Expected speedup $S$ as a function of $\delta$ for several acceptance rates $\alpha$ (shaded green indicates $S > 1$).

non-negligible $\delta$, the denominator grows further, making dense-tree drafting not just ineffective but counterproductive.

Accordingly, the success formula must be revised for large-batch inference. Feasible acceleration requires keeping $\gamma$ small so that $\tau$ remains high, minimizing $\delta$ through efficient drafter design, and increasing $\lambda$ via accurate, high-confidence predictions rather than by flooding the verifier with options. In other words, speculative decoding must shift from a breadth-oriented paradigm to an accuracy-driven one: real gains come not from generating many speculative paths, but from proposing a compact sequence that is both fast to verify and likely to be accepted.

## 4.2 EVALUATING EXISTING DRAFTER ARCHITECTURES

We evaluate three representative drafter families, standalone drafters, predictive-head drafters, and self-speculative methods, under the large-batch constraints identified in Sec. 4.1. Figure 2 summarizes empirical factors that map directly to the quantities in equation 3.1.

**Standalone drafters.** A smaller, independently deployed model avoids architectural coupling but incurs two penalties at large batch sizes (Fig. 2, left). First, the drafter's forward latency remains a sizable fraction of the target's: $\delta = 0.56$ (batch 64), $0.50$ (128), and $0.46$ (256). These values are far from the near-zero regime required for substantial speedup. Second, maintaining a separate KV cache consumes considerable VRAM that scales linearly with the batch size (e.g., $7 \rightarrow 28$ GB for 1.7B; $9 \rightarrow 36$ GB for 8B), directly constraining the usable context length and batch capacity. The combination of nontrivial $\delta$ and additional memory pressure renders the standalone design difficult to justify under large-batch inference, except possibly when pairing an extremely large target (e.g., Llama-3.1-405B (Grattafiori et al., 2024)) with a sufficiently smaller drafter.

**Predictive-head drafters.** Lightweight heads attached to the target model achieve very low $\delta$, but their reliance on tree drafting is misaligned with large-batch constraints. Figure 2 (middle) shows that $\tau(\gamma)$ collapses as both $\gamma$ and batch size increase. To mirror common tree budgets in prior predictive-head systems (Li et al., 2024b; 2025b), we evaluate $\gamma = 40$ and $\gamma = 60$. At batch 256, $\tau(40) = 0.21$ and $\tau(60) = 0.15$, versus 0.98 at batch 1. Thus, widening the tree to raise $\lambda$ rapidly pushes verification into a regime where $S \leq 1$ even under optimistic acceptance. Moreover, extending tree drafting and verification to large-batch operation is non-trivial in practice, as noted by prior systems reports on productionizing speculative decoding and surveys of batching challenges (Tang et al., 2025; Ryu & Kim, 2024; Cai et al., 2024). That said, if predictive-head drafters attain sufficiently high draft accuracy to operate with *chain* drafting (small $\gamma$), their very low $\delta$ makes them the most promising route to large-batch acceleration.

**Self-speculative decoding.** Self-speculation reuses a partially executed variant of the target (via layer skipping, early exit, quantization, or KV cache modifications), which typically yields stronger alignment and higher acceptance than lightweight heads (Zhang et al., 2024; Elhoushi et al., 2024; Xia et al., 2025; Sadhukhan et al., 2025; Tiwari et al., 2025). The drawback is that shared computation keeps the $\delta$ high: a substantial fraction of the target's path is still executed, so $\delta$ remains large, and even optimistic acceptance cannot compensate. Consistent with our sweep in Fig. 2 (right), when $\delta \in [0.5, 0.9]$ the system is at or near break-even unless acceptance is very high (e.g., at $\delta = 0.7$ one

needs acceptance $\alpha \geq 0.90$ just to reach $S \approx 1.02$), and achieving $S \geq 1.2$ generally requires $\delta \leq 0.6$ together with very high acceptance rate $\alpha$.

Within this family, layer-skipping approaches reduce cost by selecting a subset of layers. SWIFT (Xia et al., 2025) reports that pushing $\delta$ toward $\approx 0.5$ relies on *on-the-fly*, per-input layer-set optimization to adapt the skipped-layer configuration during inference, which complicates large-batch operation where homogeneous execution is essential. A separate line compresses or quantizes the draft KV cache (Sadhukhan et al., 2025; Tiwari et al., 2025), which is most effective in long-context regimes where KV traffic dominates. In aggregate, these characteristics make the $\delta$ range we evaluate (0.5-0.9) realistic for self-speculation, and they constrain large-batch viability: unless one can simultaneously drive $\delta$ low *and* maintain high acceptance without per-request adaptivity, substantial speedups are unlikely.

### 4.3 A Promising Candidate for Large-Batch Speculative Decoding

Motivated by these limitations, we investigate whether any existing mechanisms that were not originally developed for speculative decoding can nonetheless satisfy the revised criteria for large-batch acceleration. Through this investigation, we identify *Multi-Token Prediction with Gated LoRA* (Samragh et al., 2025) as a particularly promising candidate, combining architectural efficiency with high draft accuracy without relying on wide tree structures.

**Multi-Token Prediction with Gated LoRA.**    Samragh et al. (2025) shows that LLMs can forecast a short horizon of future tokens by interleaving *regular* tokens $x$ with special *mask* tokens $m$, enabling multi-token prediction (MTP) in a single forward pass. We adopt their speculative-decoding view. At each step, the input is formed by grouping every regular (anchor) token with $k$ following masks so that each mask group predicts the next $k$ futures conditioned on its immediately preceding anchor. For instance, with $k{=}2$ and anchors $(x_0, \hat{x}_1, \hat{x}_2)$, the input is $[x_0, m, m; \ \hat{x}_1, m, m; \ \hat{x}_2, m, m]$. Drafting and verification are co-located via *Gated LoRA*: in LoRA-enabled layers a binary gate $g$ is applied per position, $Y = XW + (g \odot X)AB^\top$, so that mask positions ($g{=}1$) traverse the LoRA path to produce MTP logits, while anchor positions ($g{=}0$) use the base weights only, preserving verification fidelity to the original target model.

If $\hat{x}_1$ is accepted and $\hat{x}_2$ rejected, the verified prefix becomes $[x_0, \hat{x}_1]$ and the base path yields the *bonus* next token $x_2$ from $\hat{x}_1$. The masks following the last accepted anchor (here, after $\hat{x}_1$) simultaneously provide the next-step drafts (namely, $\hat{x}_3, \hat{x}_4$), so the following step proceeds with anchors $[x_2, \hat{x}_3, \hat{x}_4]$ and the same layout. In general, each step builds an input of length $(k{+}1)^2$ (where $k{+}1$ anchors, each followed by $k$ masks), hence both verification of current drafts and drafting of future proposals happen in a single forward pass.

**Why Is It Suitable for Large-Batch Speculative Decoding?**    This approach exhibits a particularly favorable trade-off for large-batch inference. By collapsing verification and drafting into a single forward pass, it eliminates the need for auxiliary target variants or multiple sequential calls. Unlike conventional self-speculative decoding methods that rely on forwarding target variants multiple times, this method employs a unified model invocation with moderately longer input sequences, resulting in significantly improved drafting efficiency. Moreover, the drafts are generated directly from the full-capacity target model, yielding high acceptance rates and aligning well with the low-tolerance regime of large-batch inference. Taken together, these characteristics position this method as a strong candidate for satisfying the practical success formula of large-batch speculative decoding.

**Our Implementation and Adaptation.**    While the original work demonstrates the conceptual feasibility of multi-token prediction with gated LoRA, it omits many key implementation details and does not release any code implementations. To operationalize this idea for speculative decoding, we develop a complete implementation of Gated LoRA MTP decoding, encompassing (i) a fine-tuning framework for multi-token prediction using gated LoRA, (ii) a verification pipeline aligned with speculative decoding semantics, and (iii) an inference path optimized for high-throughput generation. Our implementation is integrated into a high-performance decoding engine built on FlashInfer, enabling scalable inference with speculative acceleration under large-batch settings. To facilitate reproducibility, the implementation will be released upon acceptance.

Table 1: Acceleration performance results comparing MTP with Gated LoRA against the standard decoding. We report the estimated end-to-end speedup factor and goodput (tokens/s) in parentheses. The maximum sequence lengths are set at 32k for batch sizes 16 to 64, 24k for 128, and 16k for 256. For each configuration, the tensor parallel size is chosen to match VRAM requirements.

| Strategy | $B$ | Qwen3-8B | | | DeepSeek-R1-Distill-Llama-8B | | |
|---|---|---|---|---|---|---|---|
| | | AIME2025 | CodeForces | GPQA-Diamond | AIME2025 | CodeForces | GPQA-Diamond |
| Greedy | 16 | ×2.61 (1088.3) | ×2.55 (1057.4) | ×2.40 (1003.3) | ×2.22 (1094.4) | ×2.14 (1056.0) | ×1.93 (954.1) |
| | 32 | ×2.64 (2031.2) | ×2.52 (1975.2) | ×2.37 (1865.2) | ×2.21 (2043.6) | ×2.11 (1953.9) | ×1.93 (1796.0) |
| | 64 | ×2.66 (3523.5) | ×2.51 (3303.8) | ×2.40 (3182.3) | ×2.32 (3630.9) | ×2.21 (3447.7) | ×2.06 (3218.0) |
| | 128 | ×2.68 (4799.0) | ×2.55 (4545.4) | ×2.46 (4427.1) | ×2.46 (5051.4) | ×2.29 (4756.9) | ×2.19 (4573.9) |
| | 256 | ×2.37 (6746.2) | ×2.22 (6319.6) | ×2.22 (6224.4) | ×2.15 (7000.3) | ×2.02 (6610.0) | ×1.98 (6503.6) |
| Sampling | 16 | ×2.26 (936.3) | ×2.11 (871.1) | ×1.98 (818.8) | ×1.98 (977.8) | ×1.80 (904.0) | ×1.68 (833.4) |
| | 32 | ×2.31 (1776.9) | ×2.08 (1610.8) | ×1.90 (1488.4) | ×1.99 (1857.5) | ×1.81 (1675.0) | ×1.70 (1583.2) |
| | 64 | ×2.32 (3026.8) | ×2.15 (2809.0) | ×2.03 (2648.4) | ×2.07 (3206.3) | ×1.94 (3004.6) | ×1.82 (2834.7) |
| | 128 | ×2.38 (4221.1) | ×2.14 (3790.3) | ×2.09 (3721.1) | ×2.22 (4548.9) | ×2.06 (4264.5) | ×2.01 (4177.1) |
| | 256 | ×2.03 (5730.2) | ×1.91 (5367.4) | ×1.88 (5314.6) | ×2.01 (6484.4) | ×1.86 (6064.3) | ×1.87 (6098.0) |

## 5 EVALUATIONS

**Setup.** We measure the acceleration performance of MTP with Gated LoRA, on Qwen3-8B (Yang et al., 2025) and DeepSeek-R1-Distill-Llama-8B (Guo et al., 2025). Our custom inference engine, built on FlashInfer (Ye et al., 2025) kernels, incorporates standard optimizations such as weight fusion, tensor parallelism, and paged KV cache, using `bfloat16` precision. To assess performance across diverse domains, we use three datasets: AIME2025 (OpenCompass, 2025) (Math), Code-Forces (Penedo et al., 2025) (Code Generation), and GPQA-Diamond (Rein et al., 2024) (Science). All measurements exclude prefill and initialization overhead. For speculative decoding, we report goodput (committed tokens/s), which is equivalent to throughput in standard decoding. In our experiments, we exclusively evaluate acceleration performance, since speculative decoding is theoretically guaranteed to preserve the output distribution of the target model.

**Evaluation Protocol.** To rigorously evaluate decoding throughput for long sequences (e.g., up to 32k tokens), we adopt a protocol designed to mitigate biases arising from variable generation lengths. Since throughput is highly dependent on the KV cache size, simply averaging over entire generations can be misleading. Instead, we measure throughput at various discrete prefix lengths, effectively capturing performance at different stages of the generation process. At each prefix length, decoding continues until 128 tokens are generated per sequence, and the process is repeated 10 times. We then aggregate these length-conditioned measurements to estimate the effective end-to-end throughput for a target sequence length. Full details of this protocol are provided in Appendix B, and full-sequence end-to-end measurements are reported separately in Appendix C.

### 5.1 MAIN RESULTS

Table 1 presents compelling evidence that speculative decoding with Gated LoRA MTP delivers substantial and consistent acceleration across a wide range of tasks and batch sizes. Throughput improvements persist across both Qwen3 and DeepSeek-R1 models, showing that the benefits are not tied to a particular backbone or implementation detail. This result bridges the gap between theoretical feasibility and practical deployment in real-world serving scenarios.

Closer inspection reveals three key patterns. First, greedy decoding consistently outperforms sampling due to longer average acceptance lengths, as deterministic drafts more closely align with top-1 predictions. Second, speedup varies across tasks; math shows the largest gains, which is likely due to domain alignment with the math-heavy training data, whereas code and science are underrepresented. This suggests that more balanced training or extended finetuning could further improve generalization. Third, speedup increases with batch size up to 128 as larger batches intensify KV cache pressure, amplifying the effectiveness of speculative decoding. The drop at batch size 256 is attributed to the fixed 16k sequence limit, which reduces overall memory traffic rather than exposing a method-level limitation.

### 5.2 IN-DEPTH ANALYSIS

Figure 3 (left) analyzes how draft length affects speedup across different batch sizes. While longer drafts generally yield better performance at small to medium batch sizes (e.g., batch 16 to 64), an

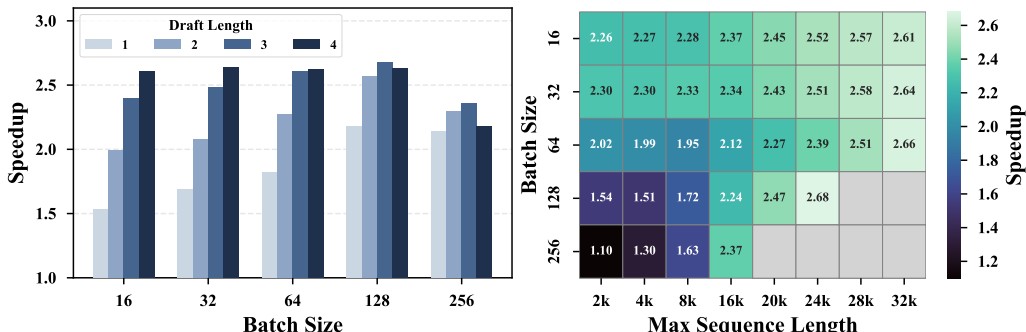

Figure 3: Ablations under greedy decoding for Qwen3-8B on AIME2025. *(Left)*: Speedup vs. draft length $\in \{1, 2, 3, 4\}$ across batch sizes. *(Right)*: Speedup as a function of batch size and maximum sequence length. Gray cells denote configurations omitted due to VRAM limits on H100$\times$8.

inverse pattern emerges at larger batch sizes. At batch 128 and 256, shorter drafts such as $k=2$ or $k=3$ actually outperform longer ones like $k=4$. This shift reflects the diminishing return of increasing draft length when the cost of forward passes grows rapidly with input length. As established in Section 3.1, larger batches significantly amplify the input-length sensitivity of decoding latency, meaning that the extra cost of verifying longer drafts can outweigh the gains from higher acceptance rates. Therefore, in high-throughput regimes, moderate draft lengths strike a better balance between parallel verification and overhead, confirming that optimal draft length is system-dependent and not monotonic with respect to speedup.

Figure 3 (right) explores how speculative decoding benefits scale with both batch size and maximum sequence length. Across all batch sizes, speedup improves steadily with longer generations, rising from 2.26 to 2.61 at batch size 16 and from 1.10 to 2.37 at batch size 256. This pattern highlights the role of KV cache pressure in inducing memory-bound behavior, making speculative decoding more effective for long-form generation tasks. Moreover, the benefits of increasing sequence length are amplified at higher batch sizes: at a batch size of 64, a 24k context is required to achieve a 2.39$\times$ speedup, whereas a batch size of 256 achieves a comparable 2.37$\times$ at just 16k. This suggests that larger batches naturally accumulate higher KV cache memory pressure, allowing them to enjoy the benefits of speculative decoding at shorter context lengths. These results not only validate our theoretical analysis but also offer actionable insights for practitioners: high speedup is most reliably obtained in regimes with large batch sizes and moderate-to-long generation lengths, especially when paired with appropriately tuned draft lengths.

## 6 Conclusion and Limitations

In this work, we have revisited speculative decoding for high-throughput, large-batch LLM inference through the lens of $\gamma$-tolerance, a practical metric capturing how verification length affects target model latency. Our analysis showed that meaningful acceleration remains attainable when verification is shallow, drafter overhead $\delta$ is small, and the expected number of accepted tokens $\lambda$ increases through accuracy rather than breadth. Guided by this understanding, we adopted Multi-Token Prediction with Gated LoRA, a method that aligns with large-batch constraints and delivered up to 2.37$\times$ end-to-end speedup at batch size 256. This result provides a concrete path toward scalable speculative decoding, connecting system behavior with algorithmic design.

While encouraging, this work has several limitations. First, the evaluation focuses on reasoning-oriented models. Although the underlying principles are generally applicable, further validation on broader model families, including general-purpose LLMs, remains an open direction. Second, the method has not been evaluated under very short sequence lengths (e.g., under 1k tokens), which are less common in modern workloads and often do not require acceleration. Third, while kernel and batch configurations were explored, closer integration with runtime components such as schedulers and memory management could further improve $\tau(\gamma)$ and enhance overall performance. We hope these insights offer a foundation for future advances in scalable, high-throughput LLM inference.

ACKNOWLEDGEMENT ON LLM USAGE

The authors acknowledge the use of LLM for linguistic assistance, specifically for editing and clarifying the phrasing of sentences and paragraphs. LLM was not involved in any aspect of the research content, including conceptual development, methodological design, or experimental design.

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
