# OpenReview forum: "Rethinking the High-Throughput LLM Inference: An Opportunity for Speculative Decoding"
_ICLR.cc/2026/Conference — ICLR 2026 Conference Withdrawn Submission_

### Official Review · Reviewer_pBaN · 2025-10-31

**Soundness:** 3
**Presentation:** 4
**Contribution:** 3
**Rating:** 8
**Confidence:** 4

**Summary:**

This paper provides a compelling, in-depth evaluation of speculative decoding's potential gains in high-throughput, large-batch-size settings. At its core, it defines $\gamma$-tolerance, which quantifies how latency scales with verification length. This concept is used to analyze the necessary conditions for successful speculative decoding in large-batch-size settings. The authors then identify the most suitable candidate, Multi-Token Prediction with Gated LoRA, which achieves the largest speedup in large-batch speculative decoding. It empirically demonstrates a 2.37× speedup over standard decoding.

**Strengths:**

Overall, I am quite happy about the paper, especially for its depth of analysis for feasible conditions of speculative decoding speedup under large batch scenarios.

1. The problem of acceleration of speculative decoding in large batch size settings is clear and impactful.
2. The proposed $\gamma$-tolerance metric is a refreshing contribution. It provides a clear tool for practitioners to evaluate whether implementing speculative decoding is worthwhile in their setup and circumstances.
3. There are many insights derived in the paper, with empirical validations in Sections 3 and 4.
4. The evaluation protocol is rigorous: It rules out the variable generation length bias through a carefully considered pipeline expanded in Appendix B.

**Weaknesses:**

1. The empirical results should be compared against a wide suite of existing speculative decoding methods (especially those suited for large batch size inferences) for more convincing results.
2. More analysis of the MTP approach, as well as discussions on its flexibility can be incorporated in the paper.

**Questions:**

1. The empirical results in the paper can be presented in a more convincing way. Currently, the results are weak as the goodput metric is only compared against standard decoding, not even vanilla speculative decoding. Other baseline comparisons, such as TurboSpec (the goodput paper) [1], MagicDec [2], and TETRIS [3], can greatly improve the credibility of the empirical results.
2. The paper lacks adequate analysis of the overhead introduced by the MTP with gated LoRA method. It is only briefly mentioned that “unified model invocation with moderately longer input sequences”. Is it possible to provide a more in-depth analysis?
3. The testing is done using a decoding engine built on FlashInfer. Can the authors also show empirical result comparisons when adapted in popular inference frameworks like vLLM and SGLang?
4. Does the MTP approach allow a flexible choice of draft length $k$ (or verification length $\gamma$)? This is important as the authors have shown that the choice of $\gamma$ on different kinds of hardware, batch size and conditions can impact performance significantly. This seems to significantly impact the practicability of the method.

***References***

[1] TurboSpec: Closed-loop Speculation Control System for Optimizing LLM Serving Goodput. In arXiv.

[2] MagicDec: Breaking the Latency-Throughput Tradeoff for Long Context Generation with Speculative Decoding. In ICLR 2025.

[3] TETRIS: Optimal Draft Token Selection for Batch Speculative Decoding. In ACL 2025.

---

### Official Review · Reviewer_U4EP · 2025-10-31

**Soundness:** 3
**Presentation:** 3
**Contribution:** 2
**Rating:** 4
**Confidence:** 3

**Summary:**

This paper revisits speculative decoding in high-throughput LLM inference and introduces a gamma-tolerance metric to reason about when speculation remains beneficial at large batch sizes. It combines empirical latency modeling with an optimized multi-token prediction with gated LoRA, showing up to major speedups.

**Strengths:**

- The paper is clearly written and well-motivated, with careful latency measurements and system-level insights.
- The gamma-tolerance metric provides an intuitive way to discuss the trade-off between speculation cost and throughput.
- Implementation details and scaling studies are well executed.

If the rebuttal addresses the questions, I can increase the score.

**Weaknesses:**

- Comparisons are narrow, limited to standard decoding and one internal speculative baseline. No evaluation against strong contemporaries like Medusa, TurboSpec, or SpecInfer.
- Novelty is quite limited as the MTP-GL design builds on established speculative decoding and multi-token prediction ideas. The contribution lies in careful empirical analysis rather than new algorithmic or theoretical innovation, but in this case, the paper requires large-scale extensive experiments demonstrating superior performance compared to state-of-the-art.
- The work lacks formal theoretical results and can be strengthened by some analytical studies.

**Questions:**

- Can you include experiments comparing your approach with other recent speculative or multi-token decoding systems?
- Is there a way to formalize or bound gamma-tolerance analytically, rather than treating it purely as an empirical ratio?

---

### Official Review · Reviewer_KarP · 2025-11-02

**Soundness:** 2
**Presentation:** 3
**Contribution:** 3
**Rating:** 4
**Confidence:** 4

**Summary:**

This paper revisits the conventional assumption that speculative decoding is ineffective at large batch sizes. Motivated by recent findings showing memory bandwidth rather than compute remains the bottleneck in large-batch inference, the authors introduce gamma-tolerance, a latency scaling metric quantifying how target model latency responds to verification length. Their empirical analysis demonstrates that speculative decoding remains feasible at large batches, though with narrower acceleration margins than small-batch settings. The key insight is that large-batch regimes demand accuracy-driven rather than breadth-oriented drafting strategies. Most existing drafter architectures (standalone models, predictive heads with tree drafting, self-speculative methods) fail under these constraints due to unfavorable accuracy-efficiency tradeoffs. The paper identifies Multi-Token Prediction with Gated LoRA as a promising approach and implements a high-performance system achieving up to 2.37x speedup at batch size 256 on reasoning workloads without requiring long contexts or target model modifications.

**Strengths:**

- Useful analytical framework: The gamma-tolerance metric provides a principled criterion for determining when speculative decoding is feasible, moving beyond vague notions of memory-bound vs compute-bound regimes. The formulation connecting system behavior to speedup potential is elegant.​​
- Practical system contribution: Achieving 2.37x speedup at larger batches (e.g., 256) more likely to be found than the small-batch scenarios in production demonstrates speculative decoding works in practice.
- Comprehensive empirical analysis: Systematic investigation across batch sizes (1-256), sequence lengths (256-16k), attention kernels (FlashAttention-2, FlashInfer). The kernel-dependence finding is particularly valuable, reconciling conflicting prior reports.​​

**Weaknesses:**

- There is a complete lack of **quantitative** comparison to related work making it completely impossible to judge if there is any practical scenario where this work outperforms existing methods. Speed-ups significantly larger than 2.5x e2e are fairly common for spec-dec, and while the large batch size is certainly a more challenging scenario, the real benefit over existing methods remains unclear. It certainly builds on a different aspect than previously focused on (before: better use of available flops in FFN, here: better data re-use in attention outweighing additional FFN costs)
- Model types: The effect of modern LLM architectures (DS-V3/V3.2/R1, gpt-oss, Qwen3-MoE, Kimi K2, Qwen3-next) with MoE, MLA, SSM layers could be evaluated much better as the evaluated scenarios with long contexts covered coding and advanced reasoning where these types of models are more popular than small 8B models without long-context prefill optimizations like MoE or long-context decoding optimizations like MLA -- in short, we are evaluating on a weird model-task combination here that exhibits known inefficiencies (like GQA's large KV cache size) that would not be present in practice.
- Task selection: Evaluation focuses reasoning & coding benchmarks -- two areas that have repetitive patterns and high predictability, and are thus particularly easy cases for SpecDec. How does the method perform on long-context reasoning based Q&A tasks involving RAG? How does it perform under distribution shifts (tuned on Task A, evaluated on Task B) such as different languages or reasoning-v-Q&A?
- The categorization of existing drafter architectures is missing on an entire category of drafters that seems highly relevant for this topic: simple text and n-gram based drafters list SSSD (simply scalable spec-dec; also focusing on larger batches), PIA (painless inference acceleration), ...
- The view of spec-dec under a system parameter like the batch size is impractical. The method's main impact is moving the Pareto frontier of the latency-throughput trade-off (the two metrics users care about as they are a user experience-v-cost trade-off); here batch size 1 has a legit place for existence and showing Pareto frontiers for different sequence lengths would be very meaningful to identify where switching from other spec-dec methods to the proposed one is meaningful, and possibly if there is a throughput "hole" in short-medium sequence lengths.

**Questions:**

1. Quantitative comparison to existing methods: Table 1 reports up to 2.37x end-to-end speedup at batch 256, but speedups significantly larger than 2.5x are fairly common for speculative decoding in other settings. How does this approach compare quantitatively to these and other large-batch speculative decoding methods on identical experimental conditions? What speedsup would they achieve on your setup (Qwen3-8B, DeepSeek-R1-Distill-Llama-8B, same batch sizes, same tasks)? Without such comparison, it remains impossible to judge whether practitioners should adopt this method over existing alternatives or if the contribution is primarily conceptual rather than of practical relevance.​

2. Model architecture evaluation: Your evaluation uses relatively small 8B models without long-context-optimized architectures. How does the method perform on modern architectures that dominate long-context reasoning workloads (DeepSeek-V3, DeepSeek-V3.2, Qwen3-MoE variants with mixture-of-experts layers, or models with Grouped Query Attention variants that have smaller KV cache requirements)? Since these models include architectural optimizations like MoE, Multi-Head Latent Attention, or efficient KV cache designs, do they still show similar bottlenecks? The reason matters: your choice of older baseline models may exhibit known inefficiencies (like large KV cache from standard GQA) not present in state-of-the-art production models (e.g., using MLA).​​

3. Task generalization and distribution shift: You evaluate only on reasoning and code generation (AIME2025, CodeForces, GPQA-Diamond), which are known to have repetitive patterns and high token predictability, making them particularly favorable for speculative decoding. How does acceptance rate and speedup degrade on long-context question-answering with retrieval-augmented generation tasks? How sensitive is performance to domain shift, such as training the LoRA on math-heavy data but evaluating on multilingual reasoning, medical QA, or conversational long-context tasks? Does a single trained predictor generalize across diverse tasks, or does task-specific finetuning become necessary?​

4. Missing drafter family: The categorization of existing drafter architectures omits an entire category of simple text and n-gram based drafters, including Simply Scalable Speculative Decoding and Painless Inference Acceleration, which seem highly relevant for large-batch settings since they avoid auxiliary models and complex tree structures. How does your method compare to these simpler approaches in terms of speedup, implementation complexity, and memory overhead? Do they represent viable alternatives that practitioners should consider?​

5. Latency-throughput Pareto frontier analysis: You frame the problem primarily as throughput at varying batch sizes, but users care about the latency-throughput tradeoff. Batch size 1 remains legitimate for latency-sensitive applications. Instead of reporting speedups at fixed batch sizes, can you provide Pareto frontiers showing the latency-throughput tradeoff for different sequence lengths? This would reveal where this method should be preferred over alternatives and identify potential throughput gaps at short-to-medium sequence lengths where practitioners might switch between speculative decoding methods. Does your method introduce a throughput hole that makes it undesirable for certain sequence length ranges?

---

### Official Review · Reviewer_qUn7 · 2025-11-04

**Soundness:** 3
**Presentation:** 3
**Contribution:** 3
**Rating:** 4
**Confidence:** 3

**Summary:**

This work revisits the core assumption of speculative decoding, namely that the inference bottleneck is primarily memory-bound. It introduces a new metric, gamma-tolerance, to quantify when speculative decoding is practically beneficial. Through extensive analysis, the paper finds that many existing model pairs fail to achieve real-world acceleration, despite theoretical gains. To address this limitation, the authors propose a multi-token prediction framework augmented with gated LoRA, which demonstrates consistent empirical improvements in both efficiency and accuracy.

**Strengths:**

1.	The paper provides a thorough examination of the core assumption underlying speculative decoding, introducing an important predictive metric that determines when SD yields genuine inference benefits.
2.	It also proposes a promising architectural direction that enables speculative decoding to remain effective across both low- and high-throughput regimes.

**Weaknesses:**

1.	An important scenario where speculative decoding (SD) often fails—and which challenges the memory-bound assumption—is in on-device applications, where computational resources are severely constrained. The paper does not investigate this setting, which somewhat limits its practical applicability and generality.
2.	While the study offers valuable engineering insights, it presents limited algorithmic innovation, thereby diminishing its agorithmic contribution. The work focuses primarily on empirical analysis and system-level optimization, without delivering substantial theoretical or methodological improvements to the core speculative decoding algorithm itself.

**Questions:**

Could you investigate the on-device scenario and validate your method on it?

---

### Note · Authors · 2025-11-27

**Comment:**

We sincerely appreciate the reviewers’ thoughtful and constructive feedback. We will incorporate their suggestions into the next revision. Thank you again for your time and effort.

**Withdrawal Confirmation:**

I have read and agree with the venue's withdrawal policy on behalf of myself and my co-authors.